# Job postings in the substance use disorder treatment related sector during the first five years of Medicaid expansion

Olga Scrivner[1]*, Thuy Nguyen[2], Kosali Simon[2,3], Esmé Middaugh[1], Bledi Taska[4], Katy Börner[1]

**1** Luddy School of Informatics, Computing, and Engineering, Indiana University, Bloomington, IN, United States of America, **2** O'Neill School of Public and Environmental Affairs, Indiana University, Bloomington, IN, United States of America, **3** National Bureau of Economic Research, Cambridge, Massachusetts, United States of America, **4** Burning Glass Technologies, Boston, Massachusetts, United States of America

⊕ These authors contributed equally to this work.
* obscrivn@indiana.edu

**Data Availability Statement:** Job Postings data are available from Burning Glass Technologies upon request. An interested researcher should send a request to info@burning-glass.com. The remaining

## Abstract

### Background

Effective treatment strategies exist for substance use disorder (SUD), however severe hurdles remain in ensuring adequacy of the SUD treatment (SUDT) workforce as well as improving SUDT affordability, access and stigma. Although evidence shows recent increases in SUD medication access from expanding Medicaid availability under the Affordable Care Act, it is yet unknown whether these policies also led to a growth in hiring in the SUDT related workforce, partly due to poor data availability. Our study uses novel data to shed light on recent trends in a fast-evolving and policy-relevant labor market, and contributes to understanding data sources to track the SUDT related workforce and the effect of recent state healthcare policies on the supply side of this sector.

### Methods and data

We examine hiring attempts in the SUDT and related behavioral health sector over 2010-2018 to estimate the causal effect of the 2014-and-beyond state Medicaid expansions on these outcomes through "difference-in-difference" econometric models. We use Burning Glass Technologies (BGT) data covering virtually all U.S. job postings by employers.

### Findings

Nationally, we find little growth in the sector's hiring attempts in 2010-2018 relative to the rest of the economy or to health care as a whole. However, this masks heterogeneity in the bimodal trend in SUDT job postings, with some increases in most years but a decrease in 2014 and in 2017, as well as a shift in emphasis between different occupational categories. Medicaid expansion, however, is not associated with any statistically significant change in overall hiring attempts in the SUDT related sector during this time period, although there is moderate evidence of increases among primary care physicians.

data underlying the results presented in the study are available from https://github.com/cns-iu/sudt-medicaid.

**Funding:** KS,OS,KB, EM, TN No number the Indiana University Responding to the Addictions Crisis Grand Challenge Grant url - https://addictions.iu.edu/responding-to-crisis/grand-challenge.html KB NIH 1R01LM012832-01A1 MyAura URL: https://projectreporter.nih.gov/project_info_description.cfm?projectnumber=1R01LM012832-01A1 1U01CA198934-01 NSF HSD URL: https://projectreporter.nih.gov/project_description.cfm?projectnumber=1U01CA198934-01 1713567 AISL https://www.nsf.gov/awardsearch/showAward?AWD_ID=1713567 The funders had no role in study design, data collection and analysis, decision to publish, or preparation of the manuscript. Burning Glass Technologies provided support in the form of data access but did not play a role in the study design, analysis, and decision to publish. Bledi Taska is a Chief Economist in Burning Glass Technologies who provided role in revising data collection quality and query design. The specific role of this author is articulated in the 'author contributions' section. Other authors were not funded by the BGT.

**Competing interests:** Author Bledi Taska is employed by Burning Glass technologies This does not alter our adherence to PLOS ONE policies on sharing data and materials.

## Conclusions

Although hiring attempts in the SUDT related sector as measured by the number of job advertisements have not grown substantially over time, there was a shift in the hiring landscape. Many national factors including reimbursement policy may play a role in incentivizing demand for the SUDT related workforce, but our research does not show that recent state Medicaid expansion was one such statistically detectable factor. Future research is needed to understand how aggregate labor demand signals translate into actual increases in SUDT workforce and availability.

## Introduction

Worldwide, the direct burden of illicit drug dependence increased to 20 million disability-adjusted life years in 2010 [1]. Examples of these illicit drugs are opioids, cocaine, amphetamines, and cannabis which have been prohibited under international drug control treaties. Opioids have substantially contributed to this increased burden of substance use disorders (SUDs) due to links to premature mortality and other adverse health outcomes. In the US, mental health and SUD together became the leading cause of disease burden in 2015, while nearly 3% of Americans aged 12 years or older reported SUDs in the same year [2].

The most effective SUD treatment (SUDT) is a combination of long-acting medications (usually methadone or buprenorphine) administered as part of a cognitive behavioral approach (such as counseling, family therapy, and peer support programs) [3]. The National Survey of Substance Abuse Treatment Services (NSSATS) reports that in 2017 there were 13,857 treatment facilities in the US with over 1,356,015 clients enrolled, representing only a 19% increase in total clients served since 2007 [4]. Opioid treatment programs (OTPs) are examples of these facilities where patients can obtain methadone. Despite increasing demand and perpetual waitlists for treatment, the supply of OTPs has remained low and constant over time, with around 1,500 approved programs in 2017 compared to 1,166 OTP programs reported in 2010 [5]. Alternatively, SUD patients can receive buprenorphine maintenance therapy from office-based providers (physicians, nurses practitioners and physicians assistants) approved to prescribe buprenorphine [3]. Lack of buprenorphine-waivered providers is prevalent; in 2016 no buprenorphine waivered providers were found in 47% of all US counties, nor in 72% of rural counties [6]. Persistent workforce barriers, leading to treatment underutilization, include insufficient education and training, burdensome regulatory procedures, lack of ability to refer patients for mental health and substance abuse counseling, burdensome reimbursement barriers, and provider stigma [5].

The SUDT workforce is deemed inadequate by almost any measure [7, 8]. Workforce shortages and barriers have played a prominent role in limiting treatment access among those suffering from SUDs [5, 9]. The services of potential benefit for the SUD population are also broader than just addictions treatment, as mental health care is a very frequent co-occurring need [10]. Thus, our empirical focus in this paper due to clinical evidence and due to data limitations discussed later, is the "mental health and substance abuse treatment" workforce as classified by the North American Industrial Classification System, which we refer to as the SUDT and related workforce or sector throughout this paper.

In the 36 states and DC, where Medicaid has expanded through the Affordable Care Act (ACA), Medicaid insurance inclusion has broadened to all non-elderly adults with income

levels beneath the benchmark of 138 percent of the federal poverty level [11]. Prior evidence suggests that Medicaid expansion has led to substantial increases in Medicaid reimbursement for SUD treatment [12–17]. Particularly, evaluating the Medicaid State Drug Utilization Database (SDUD) from 2011 to 2014, Wen et al. established that a 70% increase in buprenorphine prescribing and 50 percent rise in associated spending had arisen as a result of Medicaid expansions; it is not yet known how these translate to increases in total use of buprenorphine [12]. Sharp et al. show that while Medicaid expansion resulted in reduced methadone utilization, both buprenorphine and naltrexone prescriptions increased, as exhibited by the 2011-2016 SDUD data [16]. State Medicaid programs also facilitate access to inpatient and outpatient treatment services such as institutions for mental disease (IMD), inpatient and outpatient detoxification, psychotherapy, peer support, supported employment, and partial hospitalization [17].

As the medical and service use of SUD treatments continues to increase following Medicaid expansion, and sources of financing now exist for comprehensive treatment of non-Rx forms as well, these increases may lead to major–yet unexamined–implications for mental health and addiction workforce demand [8]. This paper examines the impacts of Medicaid expansion on job openings in the SUDT and related sector and investigates the nature of job openings in terms of occupations using data on the near-universe of 2010-2018 online US job postings collected by Burning Glass Technologies (henceforth BGT). While BGT has proved useful in the labor economics literature to study the effects of major policies, such as state minimum wage laws on labor demand [18], it has thus far not been used to study the SUDT workforce. BGT represents a valuable resource for this topic since typical labor data sets such as Bureau of Labor Statistics (BLS) products are available only with 2-3 year lags, while the addictions crisis is fast moving. The BGT data we use extends to the end of 2018, allowing us to examine recent trends in the sector.

Our approach takes advantage of standard difference-in-difference (DD) designs used in Medicaid expansion literature by comparing job postings between Medicaid expansion and non-expansion states before and after expansion. We test the hypothesis that insurance availability increases hiring attempts in the SUDT related sector. Specifically, we extract SUDT-related job openings and aggregate data to the state by time level. We then compare the number of online job postings in Medicaid expansion and non-expansion states from 2010 through 2018, testing for changes in the relationship after expansion. The findings from this exercise provide evidence on whether a large area of recent insurance policy has detectable effects on hiring attempts in the SUDT related workforce and thus has implications for SUDT access.

## Materials and methods

### Datasets

**Medicaid.**  Our analyses center on comparison between states that expanded Medicaid by the end of our study period (33 by 2018) versus non-expansion states (18). Medicaid expansion status information comes from the Kaiser Family Foundation [11].

The 33 expansion states are: AK, AZ, AR, CA, CO, CT, DE, DC, HI, IL, IN, IA, KY, LA, MD, MA, MI, MN, MT, NH, NJ, NM, NY, ND, OH, OR, PA, RI, VT, WA, WV, and WI. All listed states had expanded Medicaid through the ACA in the first quarter of 2014 with the following exceptions: Michigan (expanded April, 2014), New Hampshire (August, 2014), Pennsylvania (January, 2015), Indiana (February, 2015), Alaska (September, 2015), Montana (January, 2016), Louisiana (July, 2016), and Wisconsin (had not formally expanded by 2018). The late expansion states were excluded in our simple mean comparison. In the regression analyses, these states were included with the actual year of expansion. In partial implementation years,

the treatment status is coded as a fraction of actual months over 12 months. For instance, the treatment status for Michigan equals 3/12 in 2014, and equals 1 in 2015 and the following years. Though not actually adopting ACA expansion, we consider Wisconsin an expansion state due to its Medicaid coverage of adults up to the federal poverty threshold. Washington D. C., Delaware, Massachusetts, New York, and Vermont are early adopters of Medicaid expansion as they already provided similar coverage to low income adults. Similar to prior work, we included these early adopters in the analysis [19]. We also conducted a sensitivity analysis on our policy coding scheme which excluded these five states [20].

The 18 non-expansion states are: AL, FL, GA, ID, KS, ME, MS, MO, NC, NE, OK, SC, SD, TN, TX, UT, VA, and WY. Medicaid expansion has been authorized for implementation in 2019 or later in five of these non-expansion states (VA, ME, ID, NE, and UT); we treat them as non-expansion states as our data period ends in 2018.

**Hiring activity.** Our primary outcome, attempted hiring activity by employers, comes from a database of online job postings curated by BGT, a labor market analytics company that scrapes, cleans, and parses online job advertisements from approximately 40,000 job boards and websites [21]. The BGT data include industry and occupation codes, geographical location, and time of job postings, among other job identifiers. In this study, we focus on the time frame between 2010 and 2018, resulting in 174 million U.S. online job postings across all sectors of the economy. Our main outcome of interest is the hiring activity in all SUDT related establishments. According to the Substance Abuse and Mental Health Service Administration (SAMHSA), SUDT establishments are defined by the type of care offered and include outpatient, residential (non-hospital), and hospital inpatient services [22]. Outpatient centers may provide ambulatory detoxification, methadone/buprenorphine maintenance, or naltrexone treatment; residential facilities may provide short- or long-term care as well as detoxification; and hospitals may offer medically-controlled and monitored inpatient detoxification and treatment. Emergency rooms, private doctors' offices, self-help groups, prison and jails are not considered treatment facilities by this definition [22]. We used 4-digit North American Industry Classification System (NAICS) codes to identify job postings associated with the aforementioned SUDT related establishments as follows: (i) Psychiatric and Substance Abuse Hospitals (6222), (ii) Outpatient Mental Health and Substance Abuse Centers (6214), and (iii) Residential Mental Health and Substance Abuse Facilities (6232). Thus, it is not possible in standard industry classifications to separate mental health and substance disorder treatment facilities. Outpatient centers and Residential facilities are further subclassified by their 6-digit NAICS code, allowing for greater precision in identifying the SUDT related jobs. We excluded the following 6-digit NAICS categories: Residential Intellectual and Developmental Disability Facilities (623210), HMO Medical Centers (621491), Kidney Dialysis Centers (621492), Freestanding Ambulatory Surgical and Emergency Centers (621493), All Other Outpatient Care Centers (621498), and Family Planning Centers (621410). In addition, we excluded any job postings unclassified at 6-digit level to prevent overrepresentation. Note that the Psychiatric and Substance Abuse Hospitals category has only one 6-digit NAICS code, thus we kept all job advertisements that were not classified at the 6-digit level.

The BGT database records there being 143,688 job vacancy postings belonging to these three SUDT related establishments (henceforth SUDT hospitals, outpatient SUDT centers, and residential SUDT facilities, respectively, for brevity) over the 2010-2018 period with a total of 232,596 job ads removed by filtering. Given that there are quality implications on SUD patient care that depend on the composition of the workforce, our analysis also explores trends in job openings by occupation. BGT classifies each job vacancy by 2,4, and 6-digit NAICS codes and by Standard Occupational Classification System (SOC) code, enabling us to document the level of hiring activities per specific occupation sought in the ads.

**Covariates.** We control for important state characteristics that may be associated with SUDT-related labor market demand and be inadvertently causally attributed to Medicaid policy: unemployment rates, state populations, median household income, opioid prescribing rates, and drug poisoning mortality rates. Data on unemployment rates are from the BLS. State population estimates and median household incomes come from the U.S. Census Bureau. Opioid prescribing rates, measured as retail opioid prescriptions dispensed per 100 persons per year, come from the Centers for Disease Control and Prevention (CDC) [23]. Drug poisoning mortality rates come from the National Center for Health Statistics and refer to the estimated age-adjusted mortality rates; they reflect the average number of drug poisoning deaths per 100,000 persons [24]. To address concern regarding potential "overcontrolling" bias we do not control for the opioid prescribing rates and mortality rates in a sensitivity analysis as these factors may be causally affected by Medicaid expansion.

## Results

### Descriptive information

Health Care and Social Assistance, classified as NAICS industry sector 62, represents 14% of the labor force [25]; this comes to 21 million of the nearly 156 million in the labor force as of 2018 [25, 26]. Given our main focus on the hiring side of the labor market, we first assess existing estimates of employer demand from standard national-level BLS data—Job Openings and Labor Turnover Survey (JOLTS). It should be noted that the publicly accessible JOLTS data is not available at the sub national level nor for 4- and 6-digit NAICS industry classification. According to JOLTS data, the healthcare sector represents nearly 14 million job postings in 2018 [27, 28]. It should be noted that JOLTS measures active job postings, that is, the same posting will also be counted in the consecutive months if the position is not filled. In contrast, BGT measures true new postings: if the same advertisement occurs in the consecutive months, it will not be counted twice—the BGT applies deduplication procedures, removing postings with the same job title, employer, and location that recur within a 2-month window [29]. Thus, the number of job postings correspond solely to the job availability (vacancy) and not to how many employees a company is hiring. The healthcare sector (62) represents 4.8 million job ads in BGT in 2018.

The health sector constitutes about 17.2% (29,968,041 job ads) of all the 2010-2018 BGT data (174,226,357 job ads). Job ads of three SUDT industries at the 4-digit level comprise approximately 1.3% of the BGT health sector, including 226,456 job ads of outpatient SUDT centers, 44,923 job ads of residential SUDT facilities, and 104,905 job ads of SUDT hospitals. Of the SUDT sector, outpatient centers, residential facilities and hospitals thus make up 60%, 12% and 28% of the job ads, respectively. This is consistent with the NSSAT 2017 survey, in which outpatient programs outnumber other treatment facilities [22]. At the 6-digit level, we obtain 22,570 job postings in the SUDT outpatient centers and 16,213 job ads related to the SUDT residential facilities. The general trend for all BGT job postings as well as for health industry specifically is uniformly upward during 2010-2018 with a decrease in 2012 and 2017 (see Fig 1A). The National Survey of Substance Abuse Treatment Services also noted the 2017 decrease in the SUDT facilities operated by private non-profit organizations or by governments, as compared to the increase in the facilities operated by private for-profit organizations [22]. Within the SUDT sector, Fig 1A shows the increase in SUDT job ads between 2010 and 2018, particularly in 2018, which corresponds to the increase in job postings in the SUDT hospitals (see Fig 1B). Residential SUDT facilities and outpatient SUDT centers are similar in their trends, showing a decrease in job postings in 2014 and 2018 (see Fig 1B). To compare these trends with the actual number of SUDT establishments, we extracted the annual counts of establishments from the County Business Patterns (CBP) for each of the SUDT industry codes

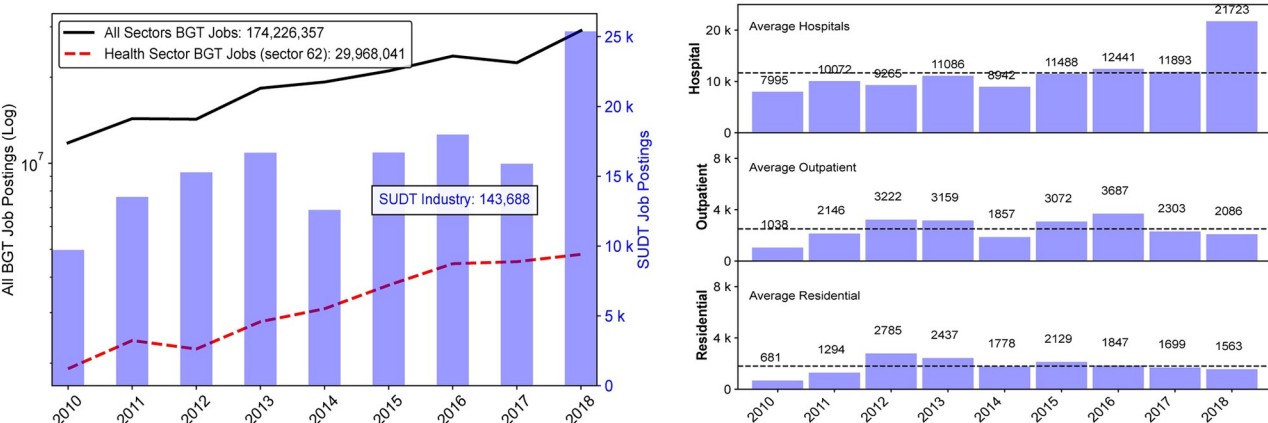

**Fig 1. BGT online job postings.** (A) BGT Job postings for all industries (black), Healthcare industry (red) and SUDT industries (light blue). The aggregated amount for all job postings is calculated for the period from 2010 through 2018. The healthcare sector is identified by the NAICS code '62'. The SUDT facilities are identified by three NAICS codes '6222', '6214','6232' filtered at 6-digit level. The left y-axis corresponds to the logarithmic trend lines for the total of all BGT job postings (black solid line) and the total of BGT healthcare sector (red dashed line). The y-right axis represents the SUDT sector values, shown as bar graphs. (B) Break down of job postings for three SUDT sectors. Three SUDT sectors are represented by their number of annual online job postings. Average line is calculated for each SUDT sector. Data Source: Burning Glass Technologies. 2019.

[30]. This CBP dataset covers the 2010-2016 period, as the 2017-2018 data is currently not available at 4-digit NAICS level. Between 2010 and 2016, outpatient SUDT centers increased on net by 1,789 to reach a total of 10,967 centers across the U.S. By 2016, residential SUDT facilities increased by 1,006 to total 7,943 establishments, whereas SUDT hospitals only added 11 new establishments nationally, to reach a total of 663 in 2016. In contrast, over the same period of time BGT outpatient SUDT centers (single physical location by employer name and 6-digit fips code) increased by 976, residential SUDT by 406, and SUDT hospitals by 1,971.

The BGT data are also unique in allowing us to track occupations specific to 4- and 6-digit NAICS industry codes. Occupation is listed in the vast majority (97% or 138,588) of all SUDT job postings; only 3.6% (5,100) are unclassified occupation job postings. Our analysis of specific occupations in the SUDT-related postings yielded 575 unique SOC occupations: 350 (outpatient SUDT), 524 (SUDT hospitals), and 291 (residential SUDT). Among the 138,588 occupation-specified job ads, we identify the following 5 most frequent occupations in this order: mental health counselors, registered nurses, medical and health service managers, psychiatric technicians, and clinical, counseling, and school psychologists. To detect any sudden increases in hiring activities, we perform a Kleinberg burst detection algorithm, a technique often used to identify unusual activities in events or novelty in terms [31, 32]. Out of 350 occupation titles for outpatient SUDT centers across 2010–2018, a total of 132 occupations displayed sudden spikes in demand and 4 occupations show double burst, for example among clinical, counseling, and school psychologists there was an increased demand in 2010 and 2018. The highest spike in workforce needs was detected for mental health counselors in 2016–2018 (Fig 2 top). This occupation, along with childcare workers, has the longest bursting demand from 2016 onward. In residential SUDT facilities, 96 out of 291 occupations exhibited a spike with 8 double-burst occupations, for example, home health aids in 2010 and 2015. The strongest hiring demands were for personal care aides in 2012–2013 (Fig 2 center). In the SUDT hospitals 192 occupations out of 524 demonstrated job postings bursting activities with 5 double bursts. The highest demand is shown in 2013 for first-line supervisors of retail sales workers (Fig 2 bottom). The bursting occupations (spikes in demands) occur for mental health counselors, registered nurses, mental health and substance abuse social workers, among others

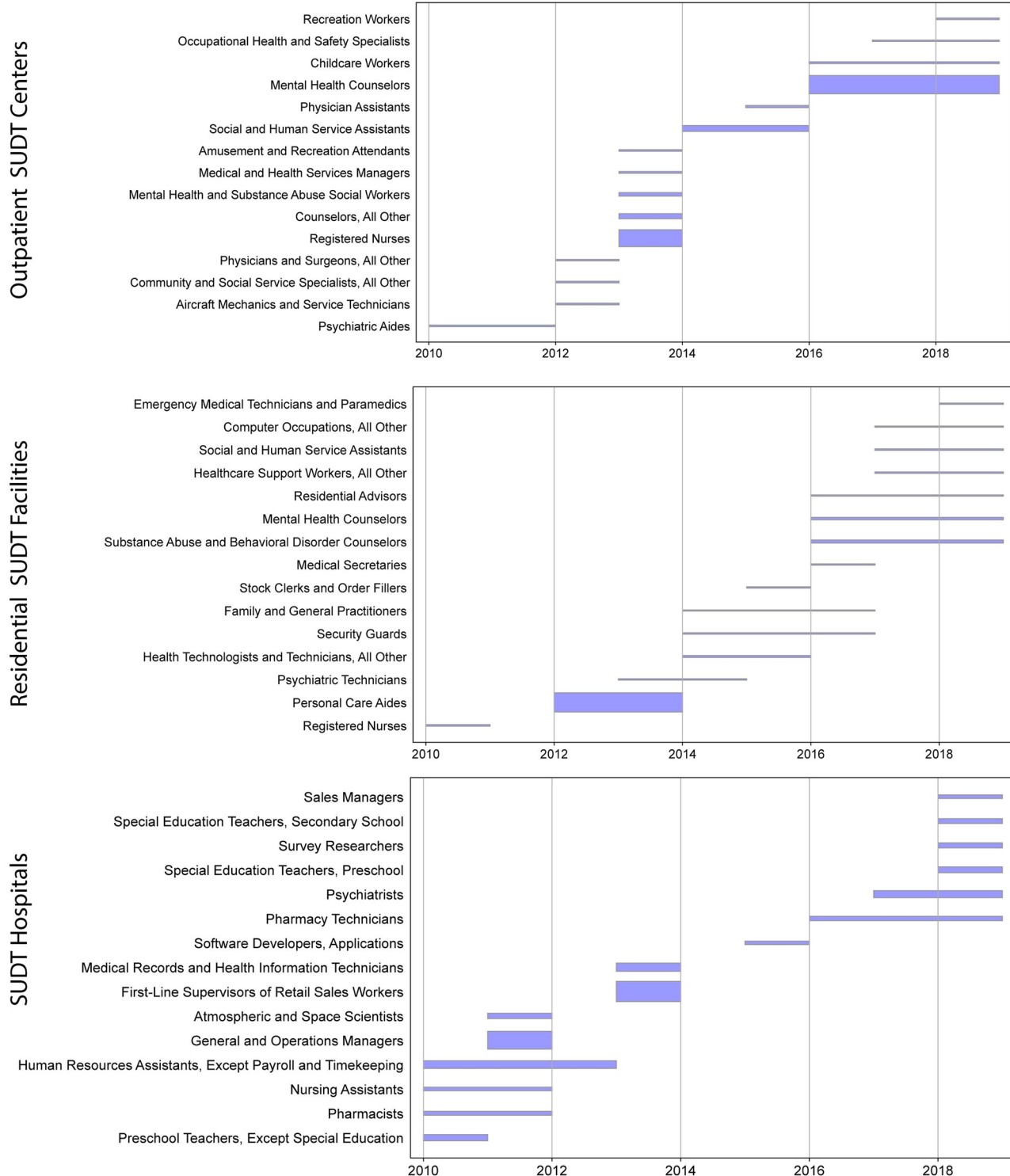

**Fig 2. Bursting top-15 SUDT occupations during 2010-2018.** Each spike in demand is shown as a horizontal bar with a start and an end date. The length of the bar corresponds to the duration of the hiring burst, the width of the bar shows the burst strength, measured as weight (e.g., in the top panel, the Mental Health Counselor occupation has the strongest and the longest burst in the years 2016–2018).

(see Fig 2). Furthermore, the burst events in the outpatient and residential SUDT sectors show a shift from registered nurses, psychiatric aides, and surgeons to mental health counselors and assistants, whereas the SUDT hospitals shift to software developers and sales managers.

## Causal analysis: Methods for estimating impact of Medicaid expansion

Our DD method essentially compares the average frequency of online US job postings for three SUDT-related industries and 457 SUDT unique occupations, in Medicaid expansion and non-expansion states, after policy change vs before. In order to comprehensively examine the effects of Medicaid expansion on hiring attempts by occupation, we grouped various SOC occupations into: (i) behavioral health professions including psychiatrists and psychologists, social workers, counselors, and therapists [33]; (ii) entry-level practitioners such as personal care aides, residential advisors, social and human service assistants, nursing assistants, and home health aides; (iii) mid-level practitioners including physician assistants, nurse practition-ers, registered nurses, and clinical laboratory technicians/technologists; (iv) advance-level, pri-mary practitioners including physicians and surgeons. These 4 groups represented 54.8% of all SUDT job postings during 2010-2018.

To identify any causal effects of Medicaid expansion on SUDT-related job postings, we draw on variation across states in adoption of Medicaid expansion in a DD empirical design. Specifically, in the Ordinary Least Squares (OLS) models, we control for: (i) state fixed effects, (ii) year fixed effects, (iii) time-variant demographic factors including unemployment rates (%) and median household income (logged), (iv) time-variant SUDT-related characteristics consisting of opioid prescribing rates (retail opioid prescriptions dispensed per 100 persons per year) and drug poisoning mortality rates. Year fixed effects are added to capture variations such as changes at the federal level, which may have affected online SUDT job postings equally across all states. State fixed effects are included to correct for unobserved heterogeneity. In par-ticular, this two-way fixed effect model (DD approach) allows us to control for all omitted state-specific time-invariant covariates that cause some states to have more job postings related to SUDT than others. Since observations in the same state may have correlated errors, we clus-ter-correct the standard errors at the state-level.

Visual inspection in S1 Fig of our main outcome distribution demonstrates a strong posi-tive skewness without an "excess zeros" issue. In particular, all states had at least one SUDT-related job posting. We used an OLS estimation which regresses the number of job postings per 100,000 state residents (which takes a logged form) on aforementioned predictors. We also conducted a sensitivity check in this estimation method through two negative binomial mod-els. These models were fit with standard regression software (Stata).

In order to evaluate the underlying assumption of the DD design in this current study–that is in the absence of Medicaid expansion, there would have been parallel trends in the control and treatment states–we present event study results. This helps evaluate whether Medicaid expansion states trends were similar to non-expansion states prior to expansion implementation. In particu-lar, we regress the number of vacancies on dummies for any pre-policy trend periods (4 years or more before expansion, 3 years before expansion, and 2 years before expansion) and dum-mies for any post periods (implementation year, and 1 to 4 years after expansion). A significant coefficient of any pre-policy trend periods may suggest a violation of the assumption underlying our DD. We use the same sets of covariates of the DD models in these event study analyses.

## Causal analysis: Regression results

Fig 3 shows the raw unadjusted job vacancy trends for the SUDT sector for Medicaid expan-sion states and non-expansion states (Fig 3A). We observe fairly consistent patterns between

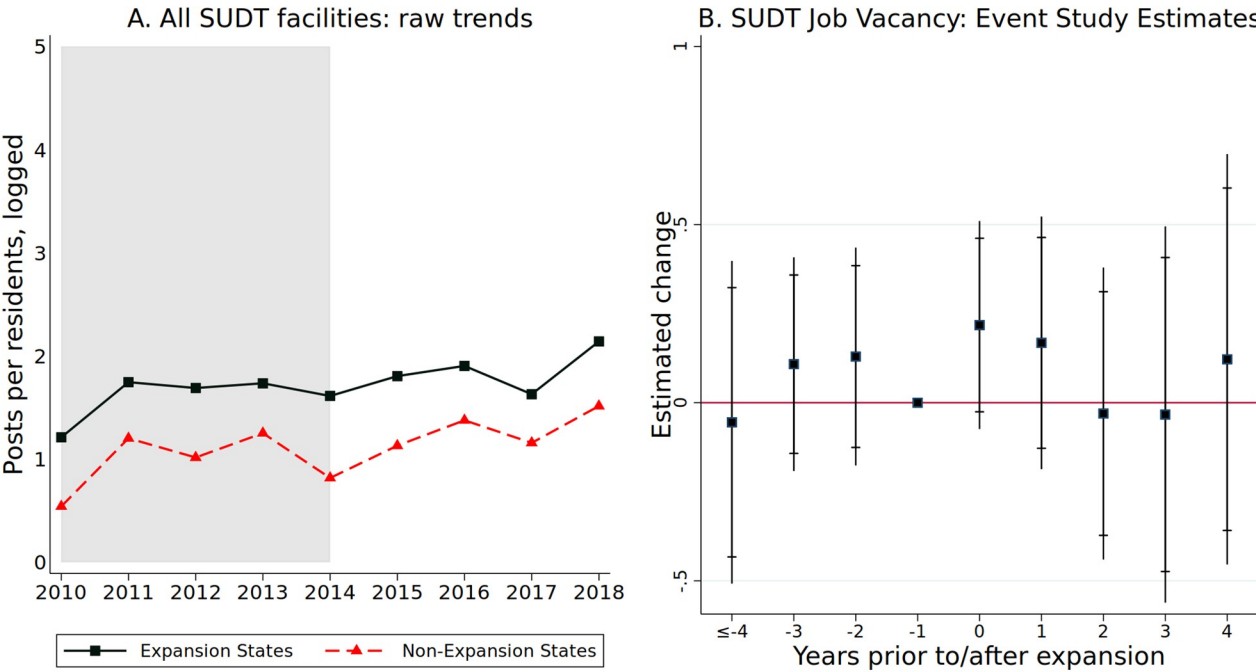

**Fig 3. Raw trends and event study estimates of job postings for SUDT.** Authors' calculations based on NAICS-state data from BGT, 2010-18, CDC prescribing rates, CDC drug poisoning mortality rates, and socio-demographic data from the BLS and Census Bureau. Panel A: we calculated the raw means of job postings per 100,000 state residents (which took log forms) for Expansion States and Non-Expansion States from state BGT data. Late expansion states (AK, IN, LA, NH, MI, MT, and PA) are excluded from this comparison. Panel B: plots the estimated difference and its 95 and 90 (bar) percent confidence intervals for each period prior to and after the implementation of Medicaid Expansion. The dependent variable is the logged number of job postings per 100,000 state residents. Late expansion states, together with 43 other states, were included in this analysis. In this event study regression, we controlled for state fixed effects, year fixed effects, median income (logged), unemployment rate, opioid prescribing rates, and age-adjusted mortality rates for drug poisoning (one- year lag values of these control variables).

the two sets of states in the pre-reform (2014) as well as in the post-reform period, which suggests that Medicaid expansion did not cause a meaningful change in SUDT sector hiring attempts. In order to understand the context better, we also examine the pattern of results for all other industries. S2 Fig, for all of the industries in the economy (except SUDT sector) and for all healthcare industry (except SUDT sector), exhibits similar parallel patterns pre and post Medicaid expansion. In particular, in S2 Fig we see parallel trends in all non-SUDT industries and healthcare sector prior to and after Medicaid expansion. S3A–S3H Fig looks at specific occupations. This analysis shows the aggregate effect across all occupations, which has a point estimate very close to zero. However, there is evidence of statistically significant increases in some of the separate occupations we study: counselors, and advanced-level primary practitioners. These comparisons provide some preliminary evidence on increases in job postings only for certain professions following Medicaid expansion but not others, which leads to an aggregate result of no increase on average [34].

A key identifying assumption of our DD model was that expansion and non-expansion states would have trended similarly in the absence of expansion. We first visually assessed trends in Fig 3A. We then formally tested for pre-policy parallel trends by examining the coefficients on the pre-expansion interaction terms in our event study model, presented in Fig 3B. The coefficients and 95 (and 90)% confidence intervals for each interaction term are plotted in this figure. This event study analysis suggests that expansion and non-expansion states were similar regarding the frequency of SUDT job ads. Using the DD and event study design,

however, we are unable to detect significant increases in the number of SUDT job ads following Medicaid expansion.

We further estimate DD models separately for three types of SUDT facilities and the most relevant seven SUDT and other occupations: (1) Outpatient SUDT, (2) Residential SUDT, (3) Hospital SUDT, (4) psychiatrists and psychologists; (5) social workers (SOC codes: Mental Health and Substance Abuse Social Workers; Child, Family, and School Social Workers; and Healthcare Social Workers, All Other); (6) counselors (SOC codes: Mental Health Counselors; Educational, Guidance, School, and Vocational Counselors; Substance Abuse and Behavioral Disorder Counselors; and Counselors, All Other); (7) Marriage and Family Therapists; Physical Therapists; Occupational Therapists; Recreational Therapists; Respiratory Therapists; Radiation Therapists; Massage Therapists; (8) Entry-level positions (SOC codes: Home Health Aides; Psychiatric Aides; Physical Therapist Aides; Pharmacy Aides; Personal Care Aides; Medical Assistants, Nursing Assistants, Therapy/Therapist Assistants, Social and Human Service Assistants; Residential Advisors; Technicians; Childcare Workers; Medical Secretaries; and Healthcare Support Workers); (9) Mid-level practitioners (SOC codes: Physician Assistants; Nurse Practitioners, Registered Nurses, Licensed Practical and Licensed Vocational Nurses, Clinical Laboratory Technicians/Technologists, Health Technologists and Technicians); (10) Advance-level primary practitioners (SOD codes: Physicians and Surgeons; Family and General Practitioners; and Health Diagnosing and Treating Practitioners); and (11) all other occupations.

Overall, the DD estimates in Fig 4 suggest that Medicaid expansion does not increase the number of job postings (all professionals, collectively) and of any particular occupation, given

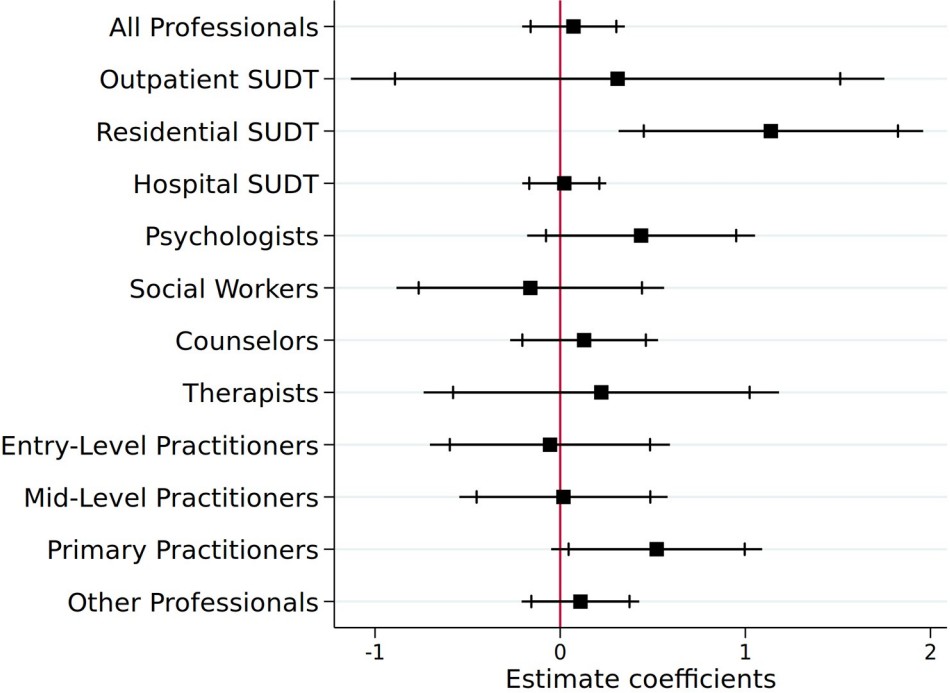

**Fig 4. DD estimates for impact of Medicaid expansion on SUDT job postings by occupation.** Authors' estimations based on NAICS-state data from BGT, 2010-18, CDC prescribing rates, CDC mortality rates, and socio-demographic data from the BLS and Census Bureau. Late expansion states (AK, IN, LA, NH, MI, MT, and PA), together with 43 other states, were included in this analysis. We use 1 year lagged values of the control variables. Standard errors were clustered at the state-level. The dependent variable is the number of Job Postings per 100,000 state residents, which takes a logged form.

the number of different specifications we run. At the 10% significance level, we observed a 68.4% increase in the ads for advanced primary practitioner. We also observed a positive association between residential SUDT job postings and Medicaid expansion, however, some pretrends (S4 Fig) may confound the comparison for this outcome. On the whole, these regression estimates do not indicate substantial impacts due to Medicaid expansion.

### Sensitivity analyses

We tested the sensitivity in our estimation method through two negative binomial models. The count outcome variable in the first negative binomial is the count of job postings by employers hiring in the SUDT workforce in a state (Model 2, S1 Table). We additionally controlled for state populations in Model 2. The count outcome variable in the second negative binomial model is the count of job postings per 10,000 residents which was rounded to an integer in this count model (Model 3, S1 Table). We used a large denominator in order to increase the accuracy of this rounding exercise. In these count models, we also controlled for state and year fixed effects. The DD estimates using negative binomial models are similar to those of the baseline model: the coefficients are positive and statistically insignificant.

In a sensitivity analysis for the main results, we do not control for the opioid prescribing rates and mortality rates as these factors may be associated with Medicaid expansion (Model 3, S1 Table). The results are similar to those of the baseline regression. We also conducted a DD analysis with one-year-lagged Medicaid expansion status as it may take time to develop an increase in the workforce size (Model 4, S1 Table). We observed no significant difference in the estimates compared to the baseline results.

We tested whether our main results are driven by a particular state by using a leave-one-out test. We conducted this test by leaving out one of 50 states and Washington D.C. in the regression process and plotted the DD estimates in S5 Fig. The results imply that there is no particular state (either a treatment state or control sate) that is extremely influential on the baseline results.

### Discussion

This paper provides a first analysis of the workforce demand side in the SUDT sector, and whether public insurance expansion is associated with a statistically significant pattern of growth. First, we note that the SUDT sector measured here as mental health and SUDT is about 5% of all healthcare sector hiring attempts in the US and that this has not increased substantially over our study time period. The lack of overall growth of SUDT job demand is unexpected, given that SAMHSA predicted increased demand for SUDT medical use and services [8, 33]. The dramatic increases in demand for SUDT services and the inadequate behavioral health workforce had been predicted following major health care reforms such as the ACA by policy makers and independent experts [8, 35]. Even though existing evidence shows a recent increase in SUD medication access, it is yet unknown whether Medicaid expansion has also led to a growth in hiring attempts in the SUDT workforce, partly due to data scarcity [33].

Using a novel data source covering virtually all U.S. online job postings by employers hiring in the SUDT workforce, we studied hiring trends in total and by top occupations within these relevant industries. Comparing the raw trends in SUDT job postings, we did not find that Medicaid expansion is associated with a visually detectable increase in SUDT job postings in the post reform period; there are also fairly consistent patterns between the two sets of states in the pre-reform (2014) period. Applying a DD design in the two-way fixed effect models, we did not detect significant increases in the number of SUDT job ads as a whole or when looking at each occupation level separately following Medicaid expansion.

This study has several limitations. First, as mentioned throughout, standard labor data classifications which use NAICS, only allow examination of the mental health and SUDT sector combined, rather than separately. However, we argue that how Medicaid expansion affects the behavioral health workforce in total is extremely relevant for SUD given strong comorbidity patterns. Since there is great need to understand the resources available for substance addictions treatment specifically, future research should find ways to better define and measure the most relevant sector for opioid use disorder treatment. Second, the findings only speak to hiring attempts: when data on actual filling of posts are released for more recent years, research should examine the effects of policy on the actual stock of employees, as our ultimate interest is in assessing adequacy of the SUDT workforce. Nevertheless, these findings are particularly relevant as some states consider changing their public insurance programs through implementation of Medicaid work requirements and other changes to the accessibility of the program, and to states that have yet to expand Medicaid. Third, several studies pointed out that online advertisements often target high-skill technical and managerial candidates, whereas blue-collar occupations are advertised off-line, affecting the representativeness of our database [27]. Additionally, online job postings may over-represent growing firms [36].

The SUDT related labor force is comprised of occupations with a variety of skill levels within clinical settings such as primary care, behavioral care, or integrated care [33]. Entry-level positions such as personal care assistants and nurse assistants often require relatively little prior training, positions for physicians and psychologists/psychiatrists require most advanced degrees (doctorate), and physician assistants, nurse practitioners, registered nurses, and clinical laboratory technicians/technologists require master's or bachelor's degree [37]. Therefore, this study further examined the effects of Medicaid expansion on reshaping the composition of the SUDT related workforce. During 2010-2018, most SUDT related hiring attempts had been made for registered nurses, medical and health service managers, mental health counselors, personal care aides, and nurse practitioners. The DD estimates suggest that expansion states tended to post more job ads related to advanced primary care practitioners online than states did not expand their Medicaid eligibility. This finding suggests compositional changes that may have clinical repercussions. These represent fruitful areas for future research to complement findings of increased use of treatment medication [12–14]. The increased hiring attempts for primary care practitioners may suggest that SUDT related establishments are recruiting a diverse workforce and integrating primary and behavioral health care. Despite prior projections that every 10% increase in the demand for SUD related treatment would result in the need for 6,800 additional SUD related counselors [8], our results suggest that although the mortality consequences of the opioid crisis continued to mount during our study period, the treatment workforce hiring attempts failed to show substantial increases; future research should continue to examine impact of alternate policy levers to provide a more comprehensive body of knowledge regarding factors that could expand the availability of treatment.

## Supporting information

**S1 Fig. Histogram of job postings in SUDT sector.** Authors' calculations based on BGT, 2010-18.
(TIF)

**S2 Fig. Job posting trends for healthcare and non-healthcare sectors.** Authors' calculations based on NAICS-state data from BGT, 2010-18. In particular, we used the NAICS-state data to compare means of job postings for Expansion States and Non-Expansion States. Estimates

were adjusted by state populations. ME and late expansion states (AK, IN, LA, NH, MI, MT, and PA) were excluded from the calculations.
(TIF)

**S3 Fig. Job vacancy trends for SUDT occupations for medicaid expansion and non-expansion states.** Authors' calculations based on NAICS-state data from Burning Glass, 2010-18. Estimates were adjusted by state populations. Late expansion states (AK, IN, LA, NH, MI, MT, and PA) were excluded from the calculations.
(TIF)

**S4 Fig. Event study estimates for SUDT job postings in residential SUDT centers.** The dependent variable is the number of job postings in residential SUDT centers per 100,000 state residents, which takes a logged form.
(TIF)

**S5 Fig. Leave-one-out analysis for the main DD result.** This figure shows the DD Estimates and their 95% CIs for Impact of Medicaid Expansion on the number of job postings per 100,000 state residents.
(TIF)

**S1 Table. DD estimates for impact of Medicaid expansion on job postings of SUDT-related industries—Robustness check with different methods.** Column 1: the dependent variable is the number of job postings per 100,000 state residents, which takes a logged form. A small amount (0.001) was added to this outcome in order to remove zeros in these analyses. Column 2: the dependent variable is the count of job postings. Column 3: the dependent variable is the number of job postings per 10,000,000 state residents, rounded to a count variable. $^*$ $p < 0.1$ $^{**}$ $p < 0.05$ $^{***}$ $p < 0.01$.
(PDF)

**S2 Table. DD estimates for impact of Medicaid expansion on job postings of SUDT-related industries—Robustness check with an alternative policy coding and different specifications.** Column 1: the dependent variable is the number of job postings per 100,000 state residents, which takes a logged form. A small amount (0.001) was added to this outcome in order to remain zeros in these analyses. Column 2: the dependent variable is the count of job postings. Column 3: the dependent variable is the number of job postings per 10,000,000 state residents, rounded to a count variable. $^*$ $p < 0.1$ $^{**}$ $p < 0.05$ $^{***}$ $p < 0.01$.
(PDF)

## Acknowledgments

The authors would like to thank Burning Glass Technologies (BGT) for data access. The authors also thank Livia Crim and Anurag Joshi for excellent research assistance. We are grateful to Jason Turi and Amanda Abraham for comments.

## Author Contributions

**Conceptualization:** Olga Scrivner, Thuy Nguyen, Kosali Simon, Esmé Middaugh, Katy Börner.

**Data curation:** Olga Scrivner, Bledi Taska, Katy Börner.

**Formal analysis:** Thuy Nguyen, Kosali Simon.

**Methodology:** Thuy Nguyen, Kosali Simon, Esmé Middaugh.

**Resources:** Olga Scrivner, Esmé Middaugh, Bledi Taska, Katy Börner.

**Supervision:** Kosali Simon.

**Validation:** Bledi Taska.

**Visualization:** Olga Scrivner, Katy Börner.

**Writing – original draft:** Olga Scrivner, Thuy Nguyen, Kosali Simon.

**Writing – review & editing:** Olga Scrivner, Thuy Nguyen, Kosali Simon, Bledi Taska, Katy Börner.

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
