## [Decision Letter · Decision Letter 0]

23 Sep 2019

PONE-D-19-19660

Hiring in the substance use disorder treatment workforce during the first five years of Medicaid expansion

PLOS ONE

Dear Dr. Scrivner,

Thank you for submitting your manuscript to PLOS ONE. After careful consideration, we feel that it has merit but does not fully meet PLOS ONE’s publication criteria as it currently stands. Therefore, we invite you to submit a revised version of the manuscript that addresses the points raised during the review process.

There is enthusiasm for this paper but also major critiques submitted by both reviewers. Reviewers provide excellent critiques of the research design, modeling strategy, and lack of discussion of the possible mechanisms underlying the expansion impact that should be addressed.

We would appreciate receiving your revised manuscript by Nov 07 2019 11:59PM. To enhance the reproducibility of your results, we recommend that if applicable you deposit your laboratory protocols in protocols.io, where a protocol can be assigned its own identifier (DOI) such that it can be cited independently in the future. For instructions see: http://journals.plos.org/plosone/s/submission-guidelines#loc-laboratory-protocols

We look forward to receiving your revised manuscript.

Kind regards,

Benjamin Cook, PhD

Academic Editor

PLOS ONE

Journal Requirements:

'The authors have declared that no competing interests exist.'

We note that one or more of the authors are employed by a commercial company: Burning Glass Technologies.

Additional Editor Comments (if provided):

Reviewers' comments:

Reviewer's Responses to Questions

**Comments to the Author**

1. Is the manuscript technically sound, and do the data support the conclusions?

Reviewer #1: Yes

Reviewer #2: Yes

2. Has the statistical analysis been performed appropriately and rigorously? 

Reviewer #1: Yes

Reviewer #2: Yes

3. Have the authors made all data underlying the findings in their manuscript fully available?

Reviewer #1: Yes

Reviewer #2: No

4. Is the manuscript presented in an intelligible fashion and written in standard English?

Reviewer #1: Yes

Reviewer #2: Yes

5. Review Comments to the Author

Reviewer #1: Please see my attached referee report. This requires 200 characters, so ................................................................................................................................

Reviewer #2: This study examines the effect of public health insurance expansions on substance use disorder treatment (SUDT) employer labor demand proxied by job postings in SUDT facilities. The analysis uses a standard DD approach for the 2010-2018 analysis period to explore the impact of Medicaid expansions on SUDT job postings in aggregate or separated by occupation. The results suggest no statistically significant effect of the public expansions on the number of SUDT job ads as a whole, but some effects when looking at occupations separately.

I have the following comments/suggestions:

1. The title is a bit misleading. Maybe the authors can think about changing the word “hiring” as the study explores job postings that might proxy employer demand, not hiring.

2. While the focus of the analysis is job postings in SUDT agencies, the introduction is focused on opioid use disorders, their burden, and treatment. While opioid use is a growing concern, especially relative to other substances, I feel that the introduction should discuss SUDs in general. After reading the introduction, I was expecting an analysis focused on OTPs and OUDs.

3. The authors use a Medicaid expansion coding scheme from the KFF. Several studies, however, follow a different coding scheme (see Miller, Wherry). According to this scheme, 5 states (DE, DC, MA, NY, VT) are considered to have adopted the expansions pre-2014. I would conduct a sensitivity test where these 5 ‘early adopters’ are excluded from the analysis sample to check if the results change.

4. While the authors explore heterogeneity by occupation and they point out that these outcomes’ distribution demonstrates a strong positive skewness, I was wondering whether a better disaggregation would be by treatment agency setting. The authors note that job postings data contains the SUDT facility setting (hospital, residential, outpatient). Heterogeneity by treatment setting might result if Medicaid plans cover more generously outpatient treatment services than, for example, residential treatment or if agencies are more likely, for any other reason, to accept Medicaid insurance for outpatient versus residential services, for example.

5. While the authors hint at increases in OUD treatment use following Medicaid expansions as responsible for changes in SUDT job ads, I think a better argument should be made on the mechanism that leads to changes in job postings after Medicaid expansions.

6. I’m not sure I understand the reasoning behind choosing an OLS model versus a model such as -nbreg- to model count data.

7. Also, why not use a Probit or logit model for dichotomous outcomes?

8. An event study (ES) is conducted to assess the parallel trends assumption. However, such a model used with more dummies for the post-expansion period (2+ years, 3+ years) is useful at uncovering model dynamics. It is possible that the effect occurs only 1-2 years post expansion or only lasts 1-2 years. In that case, the results as they are, where only one post-expansion dummy is used, are mudded.

9. I would also try to lag the Medicaid expansion variable to allow for the effect to be felt by agencies and to allow for the need for additional workforce to develop.

10. Finally, in the second part of the two part model, the sample should be restricted to state/year observations with positive numbers of job postings.

6. PLOS authors have the option to publish the peer review history of their article (what does this mean?). If published, this will include your full peer review and any attached files.

Reviewer #1: No

Reviewer #2: No

---

## [Author Response · Author response to Decision Letter 0]

22 Nov 2019

Updated Funding Statement:

BGT provided support in the form of data access but did not play a role in the study design, analysis, and decision to publish. Bledi Taska is a Chief Economist in Burning Glass Technologies who provided role in revising data collection quality and query design. The specific role of this author is articulated in the ‘author contributions’ section. Other authors were not funded by the BGT. 

Updated Competing Interest Statement

This does not alter our adherence to PLOS ONE policies on sharing data and materials.

Updated Data Sharing Statement

Burning Glass Technology data was obtained free of charge via a Data Usage Agreement between Indiana University and Burning Glass Technologies (a third party data provider) in January 2019. Job Postings data are available from Burning Glass Technologies upon request. An interested researcher should send a request to info@burning-glass.com. Other data are available via public github https://github.com/cns-iu/sudt-medicaid

All reviewer comments are addressed in Response to reviewer Letter

---

## [Decision Letter · Decision Letter 1]

15 Jan 2020

Job postings in the substance use disorder treatment workforce during the first five years of Medicaid expansion

PONE-D-19-19660R1

Dear Dr. Scrivner,

We are pleased to inform you that your manuscript has been judged scientifically suitable for publication and will be formally accepted for publication once it complies with all outstanding technical requirements.

With kind regards,

Benjamin Cook, PhD

Academic Editor

PLOS ONE

Additional Editor Comments (optional):

Reviewers' comments:

Reviewer's Responses to Questions

**Comments to the Author**

1. If the authors have adequately addressed your comments raised in a previous round of review and you feel that this manuscript is now acceptable for publication, you may indicate that here to bypass the “Comments to the Author” section, enter your conflict of interest statement in the “Confidential to Editor” section, and submit your "Accept" recommendation.

Reviewer #1: All comments have been addressed

Reviewer #2: All comments have been addressed

2. Is the manuscript technically sound, and do the data support the conclusions?

Reviewer #1: Yes

Reviewer #2: Yes

3. Has the statistical analysis been performed appropriately and rigorously? 

Reviewer #1: Yes

Reviewer #2: Yes

4. Have the authors made all data underlying the findings in their manuscript fully available?

Reviewer #1: Yes

Reviewer #2: Yes

5. Is the manuscript presented in an intelligible fashion and written in standard English?

Reviewer #1: Yes

Reviewer #2: Yes

6. Review Comments to the Author

Reviewer #1: The authors have done a nice job responding to the comments. (I have to meet 100 characters, so....)

Reviewer #2: (No Response)

7. PLOS authors have the option to publish the peer review history of their article (what does this mean?). If published, this will include your full peer review and any attached files.

Reviewer #1: No

Reviewer #2: Yes: Ioana Popovici

---

## [Editor Report · Acceptance letter]

22 Jan 2020

PONE-D-19-19660R1 

Job postings in the substance use disorder treatment related sector during the first five years of Medicaid expansion 

Dear Dr. Scrivner:

I am pleased to inform you that your manuscript has been deemed suitable for publication in PLOS ONE. Congratulations! Your manuscript is now with our production department. 

With kind regards,

on behalf of

Dr. Benjamin Cook 

%CORR_ED_EDITOR_ROLE%

PLOS ONE